Evaluation of non-invasive hair snares for North American beavers (Castor canadensis): placement, efficiency, and beaver’s behavioral response

Freund Dani R. danirfreund@gmail.com 1 2
Bump Joseph K. 2
1 Department of Environment and Life Sciences, Trent University , Peterborough , Ontario , Canada
2 Fisheries, Wildlife and Conservation Biology, University of Minnesota , St. Paul , MN , United States of America
Stefen Clara
Electronic publication date: 2025 Apr 3
Publication date: 2025
Volume: 13
Electronic Location ID: e19080
Received 2024 Oct 21; Accepted 2025 Feb 10
Copyright: ©2025 Freund and Bump
Copyright year: 2025
Copyright holder: Freund and Bump
License: This is an open access article distributed under the terms of the Creative Commons Attribution License, which permits unrestricted use, distribution, reproduction and adaptation in any medium and for any purpose provided that it is properly attributed. For attribution, the original author(s), title, publication source (PeerJ) and either DOI or URL of the article must be cited.
License URL: https://creativecommons.org/licenses/by/4.0/

Keywords: Passive-sampling, Non-invasive methods, Beavers, Hair, Animal welfare, Castor canadensis, Behavioral ecology, Hair snares, Bycatch

Funding: The University of Minnesota and the Minnesota Environment and Natural Resources Trust Fund (ENRTF) The National Science Foundation Graduate Research Fellowship Program (NSF GRFP) grant no. 2237827 Funding was provided by the University of Minnesota and the Minnesota Environment and Natural Resources Trust Fund (ENRTF). Dani R. Freund received support from the National Science Foundation Graduate Research Fellowship Program (NSF GRFP, grant no. 2237827). The funders had no role in study design, data collection and analysis, decision to publish, or preparation of the manuscript.

==============================
Although the commercial demand for North American beaver (Castor canadensis) hair shaped much of the socio-ecological landscape of North America, use of beaver hair in wildlife research has focused on the Eurasian beaver (Castor fiber) and collection methods have largely involved handling animals alive or sampling dead animals. In 2022 and 2023, we tested the utility of barbed-wire hair snares to non-invasively collect hair from beavers around ponds in Northern Minnesota. At 56 different beaver ponds, we deployed 64 hair snares with remote cameras. From these data, we determined the efficiency of hair snares to collect samples, from what side of the body samples are collected, the weight and dirtiness of samples collected, the potential for bycatch, and if snares impede beavers’ ability to travel on land. We collected beaver hair samples from 94% of snares deployed, with snares sampling beaver legs and back most often. Forty-two percent of samples collected had no dirt on them, and the most productive snare collected on average 3.4 mg of clean hair per day. Muskrats were the second most sampled animal, but only made up on average 16% of total samples recorded on video per snare. Snares inhibited beaver travel in 0.1% of videos (n = 5,627 videos of beavers recorded, n = 6 videos where beaver travel was inhibited). We did not find any predictive variable that influenced the collection of beaver hair (e.g., location of snare at pond, presence of wire brushes on snare, number of times beavers touched snares, or location on the beaver’s body that was sampled). Our study provides in depth evidence of passive hair snare methods used to collect North American beaver hair, and serves as a guide to non-invasive hair snaring for multiple objectives such as hormone, genetic, and stable-isotope sample collection.

Introduction

Hair snaring is a non-invasive monitoring technique that uses various devices to passively collect hair from wild animals. A menagerie of sampling devices have been tested, including carpets with nails and barbed wire enclosures (Magioli et al., 2019). Hair snares can be relatively simple and affordable to construct, providing a data collection method that can be scaled up to large monitoring programs. Hair itself can provide valuable information on wild populations including presence-absence (McDaniel et al., 2006), genetic structure (Patkó et al., 2016), home range size (Davoli et al., 2013), density (Gardner et al., 2010), relatedness (Woods et al., 1999), hormonal state (Koren et al., 2002), and diet (Darimont & Reimchen, 2002). Collecting hair via hair snares has been particularly successful in gaining invaluable knowledge of elusive species from the wolverine (Gulo gulo) (Lukacs et al., 2020) to the American pika (Henry & Russello, 2011).

Although the demand for North American beaver (Castor canadensis) hair (fur) has shaped much of the socio-ecological landscape of North America through indigenous use from time immemorial to the later fur trade that drove colonization (Dolin, 2010), wildlife scientists have yet to routinely capitalize on beaver hair as a tissue to gain knowledge about the species. Use of beaver hair in wildlife research has largely focused on the Eurasian beaver (Castor fiber), and collection methods have mostly involved handling animals alive or sampling dead animals (Ducroz et al., 2005; Frosch, Haase & Nowak, 2011). The few previous studies that have assessed the efficacy of hair snares (Herr & Schley, 2009; Sobkowiak, Kochan & Kruszyński, 2021) or used hair snares to collect beaver samples (Frosch et al., 2014) have been conducted on Eurasion beavers. Furthermore, such studies assumed barbed wire hair snares do not negatively impact animals due to the absence of blood on snares (Herr & Schley, 2009; Frosch et al., 2014; Sobkowiak, Kochan & Kruszyński, 2021). There is therefore a lack of detailed evidence assessing the impact snares have on beavers including influence on movement behavior (e.g., restricting travel).

We placed remote cameras on hair snares at active beaver ponds in northern Minnesota, USA to thoroughly examine hair snare application at a larger scale than, to our knowledge, has been attempted on North American beavers. From the hair samples and videos we collected we aim to: (1) Determine the efficiency of hair snares to collect samples from beavers by testing different sites for snares around beaver ponds, as well as various lures and snare attachments; (2) Determine the quality of samples collected (weight and dirtiness) as well as the potential for bycatch from species with hair similar in appearance as beavers, such as muskrat (Ondatra zibethicus), otter (Lontra canadensis), fisher (Pekania pennanti), mink (Neogale vison), and other weasels (Mustelidae spp.); (3) Evaluate how beavers react to hair snares by determining what side of the body samples are collected from and if snares impede beavers’ ability to travel and forage, or change their direction of travel. Our study provides detailed evidence of the capacity of hair snares to collect samples, as well as evaluates welfare considerations that wildlife researchers may take into account when employing hair snares as a non-invasive method.

Materials & Methods

Hair snare deployment

We deployed 64 hair snares at 56 different beaver ponds from September to October across 2 years: 33 snares in 2022 and 31 snares in 2023 (Fig. 1). We set out hair snares for this research per the University of Minnesota’s Institutional Animal Care and Use Committee (protocols: MWR_VOYA_WINDELS_WOLF and UMN protocol no. 2207-40241A) and the State of Minnesota Department of Natural Resources Division of Fish and Wildlife (Special Permit No. 35003). At five of the 56 ponds, we deployed snares in both 2022 and 2023 because they either did not collect samples in 2022 at all or the sample weights were small. Other than these five ponds, we deployed snares at different ponds in 2022 and 2023 to increase our sample size across our study area.

Figure 1 Locations of barbed wire hair snares (points, N = 64) deployed at 56 beaver (Castor canadensis) ponds in Northern Minnesota, USA between September and October in 2022 and 2023.

We deployed all snares outside the national park boundary.

We collected hair samples to ultimately better understand wolf-prey ecology, so we placed snares at ponds based on known wolf territories in the area. In northern Minnesota, beaver density is high (>0.47–1.0 colonies/km2; Johnston & Windels, 2015), and multiple beaver colony territories can occur within the same interconnected pond network (e.g., connected ponds). In order to limit the potential of sampling beavers from multiple colonies, we only deployed one snare per pond network.

To improve the chance of collecting samples, we placed snares where we observed the freshest and most active beaver sign. We specifically chose where to place snares by walking along and surveying the perimeter of each pond and assessing the presence of fresh mud, as well as freshly cut and gnawed trees and cut herbaceous vegetation. In order to determine the most productive sites for hair snares, we categorized the beaver feature at which we placed each snare as either on dam, over dam crossing, over feeding trail, over feeding area, or below lodge (Fig. 2).

Figure 2 Beaver (Castor canadensis) features where we deployed barbed wire hair snares (N = 64) between September and October in 2022 and 2023 in Northern Minnesota, USA.

Barbed wire was held up by T-posts and a remote camera recorded videos of animal activity at each snare. Image A (dam)—Snare is placed across beaver dam but there is no clear sign that beavers are crossing the dam. There is evidence beavers are spending time on the pond side of the dam from fresh mud, beaver tracks, or freshly cut vegetation; Image B (dam crossing)—Snare is placed across a clear trail with beaver tracks along it running perpendicular to the dam that goes to another water body (e.g., pond or wetland) Image C (feeding trail)—Snare is placed across a trail leading away from the beaver pond that is used to collect or consume vegetation. Active feeding trails are identified by a visible path that is cleared or has crushed vegetation, with fresh beaver cuttings along the trail that are at least over 1 m from the water’s edge; Image D (feeding area)—Snare is placed in an area next to the shore of a beaver pond where beavers are actively cutting vegetation but there is no clear path going from the water to the fresh cuttings; Image E (lodge)—Snare is placed atop or below a beaver lodge where there is fresh mud and beaver tracks; Image F—beaver traveling underneath snare placed on a feeding trail as defined in image C.

Hair snare set up

We built snares following the methods of Herr & Schley (2009) and Sobkowiak, Kochan & Kruszyński (2021). For each snare we attached three meters of four-pronged barbed wire to two T-posts that held the barbed wire loosely strung above the ground (Fig. 3). We set snares approximately 30 cm above the ground with the intention that beavers would travel underneath the snare.

Figure 3 Close up images of different hair snare methods used to collect beaver (Castor canadensis) hair non-invasively in Northern Minnesota, USA in 2022 and 2023.

(A) A close-up of 4-pronged barbed wire used as a hair snare. (B) A close-up of a steel pipe brush attached to the barbed wire of a hair snare. Both images show beaver hair on the barbed wire and pipe brush.

To determine efficient methods for collecting samples, we tested a number of different variations of the barbed wire snare. To increase the surface area that could potentially collect hair, in 2022 we attached three stainless steel pipe brushes (Saim Tube Cleaning Brush Stainless Steel Pipe Brush wire diameter: 20 mm/0.8”; bristle brush length: 12 cm/4.7”; total length: 30 cm/11.8”) on 32 of the 33 snares deployed (Fig. 3). We also tried a number of different lures to entice beavers to come to the snare. In 2022 we baited three snares with apples placed one foot beyond the snare away from the pond edge because previous research used suspended apples to bait beaver traps (Sobkowiak, Kochan & Kruszyński, 2021). In 2023 we wrapped aspen leaves and sticks around the barbed wire of 30 snares when we initially deployed them. That same year we also baited six snares with beaver chewed logs of varying species that we found around the pond by placing the logs away from the pond edge 30 cm beyond the snare. All bait was handled with bare hands.

To better understand how beavers react to the snares and record from which side of the body hair is sampled, we placed remote cameras on the majority of hair snares (Recon Force Elite HP5, Browning Trail Cameras, USA). We programmed the camera to record a 20-second video (1,920 × 1,080 p) when activated. We set video quality to “ultra” (60 frames per second), motion detection to “long range” (30.48 meters), trigger speed to “fast” (0.1 seconds), infrared flash power mode to “long range” (39.62 meters), smart infrared video to “off”, and SD management to “off”. We did not have enough cameras for all of the snares in both years so cameras were sometimes moved between snares depending on the hair samples collected and the capacity of the team (see results for the number of days we deployed cameras on snares). We recorded the species of animal captured in the video, time and date video occurred, behavior (see supplementary material for description of behaviors), age class (neonate or adult), number of individuals, if the animal touched the snare and the number of times the snare was touched, if the animal traveled over or under the snare, and what part of the body the snare touched (snout, head, neck, back, side, belly, leg, rump, tail). If the animal touched the snare we recorded it as a potential event that collected a hair sample. This is because we could not accurately tell if a hair sample was left behind from the videos, and each time the snare was touched there was a possibility that hair was left behind. We recorded if hair snares alter beaver behavior or inhibit their ability to travel on land by noting when beavers investigated snares by sniffing or chewing the barbed wire, as well as if beavers got stuck on the snare. If beavers reversed their direction of travel after investigating or getting stuck on the snare, we noted that as inhibition of travel caused by the snare.

Sample collection

We attempted to check snares every week given available resources. We removed beaver hair on snares using clean pliers and placed the hair in a dry coin envelope. We left envelopes open in a dry and dark room for 24 h after collection for samples to dry before we sealed the envelopes shut. Only beaver hair was collected, but we noted if other species of hair was on the snares in 2023. We gave samples a categorical dirt score (0: no dirt, 1: small flecks of dirt one mm or smaller, 2: clumps of dirt larger than one mm, 3: clumps of dirt larger than one mm and clumps of leaves, 4: dried mud) and weighed. We noted the number of guard hairs in every sample. If more than 10 guard hairs were present, the sample was classified as having >10 guard hairs. We noted that hair follicles were present if at least one guard hair in the sample had a follicle.

Analysis

We assessed three models to determine if any of the measured factors influenced hair collection for snares where we deployed cameras. These models tested the probability of a sample being collected using logistic regression (model 1) and if specific factors influenced the weight (mg) of hair collected (models 2 and 3) (Table 1). We completed all analyses in RStudio (version 2023.12.1 + 402 (2023.12.1 + 402)). To run the logistic regression (model 1), we used the glm function with the binomial family and the default logit link. For the linear models (models 2 and 3) we used the lm function. Model selection was not used to assess variable influence on response variables, rather, we chose predictive variables a priori based on expected ecological relevance to our response variables. We assumed predictive variables significantly influenced response variables if p-values were < 0.05. Here we define “sampling period” as the days between when we checked each snare (e.i., snare deployment or the available days to collect hair). We determined the number of times animals were sampled at snares from remote camera videos of animals coming into contact with the snare during each sampling period. We scaled and centered all predictive count variables to control for differing scales.

Table 1 Summary of models assessed to determine the efficiency of barbed wire hair snares to passively collect beaver (Castor canadensis) hair in Northern Minnesota, USA.

Sampling periods are the number of days between visits to snares to collect samples. All predictive variables beginning with “number” (e.g., count variables) were scaled and centered around the mean.

Question	Specified model	
Model 1: What influences the likelihood of a beaver hair sample being collected?	(For all sampling periods where videos were recorded at snares by remote cameras)
beaver hair sample collected ∼
brushes present +
beaver feature +
number of times beavers sampled +
number of times non-beaver animals sampled	
Model 2: What influences the weight of beaver hair samples collected?	(For all sampling periods where a beaver hair sample was collected with a dirt score of 0 and videos were recorded at snares by remote cameras)
beaver hair weight / sample days ∼
brushes present +
beaver feature +
number of times beavers sampled +
number of times non-beaver animals sampled	
Model 3: Do certain areas of the beaver body leave more hair on the snare than others when they come into contact with the snare?	(For all sampling periods where a beaver hair sample was collected with a dirt score of 0 and videos were recorded at snares by remote cameras)
beaver hair weight / sample days ∼
number of times beaver back sampled +
number of times beaver belly sampled +
number of times beaver side sampled +
number of times beaver snout sampled +
number of times beaver tail sampled	

We could not clean samples collected at the time of this analysis because they were collected for chemical analyses that would have been affected. Consequently, for models 2 and 3, we only used hair samples with a dirt score of 0 to ensure we weren’t overrepresenting the weight of samples due to the presence of dirt. The beaver feature “below dam” was not used in model 2 or 3 because no snares placed below dams collected samples with a dirt score of 0. For the response variable in models 2 and 3, we divided the weights of each sample collected by the length of the sampling period for that respective sample. By standardizing hair weight by the length of the sampling period we can determine if any other factors other than length of sampling period influenced hair collection. To control for multicollinearity in the predictive variables in model 3, we summed the number of times the head, neck, back, and rump were sampled and the number of times the belly and legs were sampled, because those groups of body parts were typically sampled together when beavers traveled under and over snares, respectively.

Results

We deployed snares for an average of 60 ± 13 sd days in 2022, and 51 ± 5 sd days in 2023. One snare was deployed across a feeding area, two below a lodge, seven at a dam, 16 at a dam crossing, and 38 across feeding trails. Sampling periods (e.g., number of days between when we checked snares and remote cameras at snares) were on average 14 ± 16 sd days and 8 ± 7 sd days in 2022 and 2023, respectively. We deployed cameras on snares for an average of 91% ± 23% sd and 86% ± 28% sd of the days each snare was deployed in 2022 and 2023, respectively. We captured 9,186 videos of animals across both years, and beavers made up on average 60% ± 25% sd and 48% ± 28% sd of the videos captured per snare in 2022 and 2023, respectively.

Beavers touched snares at least one time in 54% of the videos that captured beavers. On average, beavers touched snares 2.06 ± 4.09 times per day while cameras were deployed. Beavers made up on average 83% ± 20% sd of the total times any animal species touched the snares. One snare at a dam crossing only had a camera deployed on it for 3 days, but recorded videos of animals touching the snare 78 times, of which beavers made up 92% of total snags (snare 29, Fig. 4). Muskrats came into contact with snares the second most, but only touched the snare at 14 snares, and the times they touched snares made up 16% ± 27% sd of times animals touched snares on average per snare. Muskrats made up 95% of the times animals touched one snare, but no beavers were observed at that snare (snare 13, Fig. 4). In 2023 we collected wolf hair from five snares and bear hair from the same snare twice. No samples were collected from snares baited with apples. At 15 of the 30 snares baited with aspen, samples were collected the next time they were checked. At five of the six snares baited with chewed beaver logs, a sample was collected the next time it was checked.

Figure 4 Percent of instances remote cameras recorded animals touching barbed wire hair snares deployed at beaver (Castor canadensis) ponds in 2022 and 2023 in Northern Minnesota, USA.

Beavers went under snares 49%, over 36%, and along the side 7% of the time they crossed snares. We were not able to tell if the beaver went over, under, or along the side of the snare 8% of the time because of visual obstructions in the video (e.g., beaver crossed snare behind log or rock, or weather made it difficult to tell). However, in these cases we could still tell that the beaver came into contact with the snare from the movement of the barbed wire and the beaver traveling on either side of the snare. The legs and backs of beavers came into contact with snares the most at 17% and 18% of the time respectively. The side came into contact with snares the least at 3% of the time (Fig. 5).

Figure 5 Percent of time each body region of beavers (Casto canadensis) (snout, head, neck, back, rump, tail, side, legs, and belly) came into contact with barbed wire hair snares deployed at beaver ponds in 2022 and 2023 in Northern Minnesota, USA.

Out of 179 total samples collected across the 64 snares, four snares did not collect any samples. The average weight of clean hair samples collected per the number of days snares were deployed was 0.37 ± 0.67 sd mg per day. The most productive snare collected on average 3.4 mg of clean hair per day. The largest sample with a dirt score of 0 was 113 mg. The average dirtiness of all samples was 1.12, with samples scoring, 0, 1, 2, 3 and 4 for dirtiness making up 42%, 28%, 13%, 10%, and 7% of all samples collected respectively. We collected at least one guard hair in 87% of samples and 50% of samples contained 10 or more guard hairs. Follicles were present on at least one guard hair in 55% of samples that collected guard hairs. Beavers investigated snares in 5% of the videos that recorded beavers, and their travel was inhibited (e.i. got caught on snare) in 0.1% of the videos that recorded beavers. In these instances, beavers released themselves when they were caught on snares in less than a second. No predictive variables in our three models significantly influenced the collection of hair at a significance level of 0.05 (Tables 2, 3 and 4).

Table 2 Results from logistic regression investigating the likelihood of a beaver (Castor canadensis) hair sample being collected (1 = collected, 0 = not-collected) from a barbed wire hair snare (n = 108 samples), in Northern Minnesota, USA.

The intercept represents the log odds a sample is collected at a snare placed below a beaver lodge without metal brushes present. Estimates of categorical variables indicate the change in log-odds of collecting a beaver hair sample from a snare placed at a dam, a dam crossing, or a feeding trail compared to a snare below a beaver dam. Estimates of standardized count variables indicate the change of log odds of collecting a hair sample with every 1-standard deviation increase of the explanatory variable. SE indicates standard error and p-values are rounded.

	Variable type	Coefficient estimate	SE	p-value	
Intercept		−16.53	2797.44	0.10	
Brushes present	Categorical	15.92	1901.07	0.10	
Beaver feature (dam)	Categorical	18.58	4043.67	0.10	
Beaver feature (dam crossing)	Categorical	1.82	3382.27	0.10	
Beaver feature (feeding trail)	Categorical	17.99	2797.44	0.10	
Number of times beavers sampled	Standardized count	1.95	0.10	0.05	
Number of times non-beaver animals sampled	Standardized count	0.31	0.70	0.66	

Table 3 Results from a linear model investigating what influences the mass of beaver (Castor canadensis) hair samples collected per day from barbed wire hair snares (n = 47 samples), in Northern Minnesota, USA.

Estimate of the intercept represents the average mass of beaver collected (mg/day) from a snare at a beaver dam without metal brushes present. Estimates of categorical variables indicate the change in the average mass (mg/day) of beaver hair collected at dam crossings or feeding trails compared to dams. Estimates of standardized count variables indicate the change in the average mass (mg/day) of beaver hair collected for every 1-standard deviation increase in the explanatory variable when snares are placed at beaver dams. SE indicates standard error and CI indicates confidence interval. P-values are rounded.

	Variable type	Coefficient estimate	CI [2.5%–97.5%]	SE	p-value	
Intercept		0.05	[−11.24–11.37]	5.60	0.10	
Brushes present	Categorical	−0.54	[−7.65–6.57]	3.52	0.90	
Beaver feature (dam crossing)	Categorical	0.81	[−8.04–9.66]	4.38	0.85	
Beaver feature (feeding trail)	Categorical	1.99	[−9.16–13.10]	5.51	0.72	
Number of times beavers sampled	Standardized count	0.57	[−0.90–2.05]	0.73	0.44	
Number of times non-beaver animals sampled	Standardized count	−0.72	[−2.30–0.87]	0.79	0.37	

Table 4 Results from a linear model investigating what influences the size of beaver (Castor canadensis) hair samples collected per day from barbed wire hair snares (n = 47 samples) in Northern Minnesota, USA.

The intercept in this model is not biologically meaningful so it is not included in this table. The estimates of each explanatory variable represent the average mass of beaver hair collected (mg/day) per 1 standard deviation increase in the number of times each body part was sampled (Head/Neck/Back/Rump, Belly/Legs, Side, Snout, Tail). Body parts that were regularly sampled together were combined in the analysis to avoid multicollinearity (Head/Neck/Back/Rump and Belly/Legs). SE indicates standard error and CI indicates confidence interval. P-values are rounded.

Sampling frequency location	Variable type	Coefficient estimate	CI [2.5%–97.5%]	SE	p-value	
Head / Neck / Back / Rump	Standardized count	1.09	[−0.92–3.10]	0.10	0.28	
Belly / Legs	Standardized count	0.44	[−1.36–2.24]	0.89	0.62	
Side	Standardized count	0.36	[−2.10–2.83]	1.22	0.77	
Snout	Standardized count	−1.24	[−4.95–2.47]	1.84	0.50	
Tail	Standardized count	−0.27	[−2.62–2.07]	1.16	0.82	

Discussion

Our study indicates that the methods followed here will lead to successful collection of high-quality beaver hair samples a majority of the time. Snares set at beaver ponds (e.g., areas of high beaver activity) successfully targeted beavers the most out of all species, and we successfully collected samples from almost all 64 snares. On average, samples were not very dirty, indicating the utility of this method to collect samples for processes that require minimum washing post collection (e.g., hormone studies where washing may leach hormones from the sample Koren et al., 2002). Furthermore, the approach tested here does not appear to negatively impact or impede beaver movements, and does not raise apparent welfare concerns since animals regularly snag hair on natural features such as rocks and branches. Our data therefore will help guide further use and refinement of non-invasive methods used for beavers.

Of the variables explored, none were identified as significant factors influencing the efficiency of snares. It therefore appears that the addition of metal wire brushes is not a necessary effort to the hair snare set up. Interestingly, the number of times animals snagged snares or the side of the body that was snagged did not significantly influence the successful capture of hair or the size of the sample if a sample was collected. This is likely because each time an animal touches a hair snare, there is not a guarantee a sample is left behind. Consequently, a large sample may come from a single decent snag of any side of the body, rather than multiple snags over time. These findings are important to consider if studies are designed to collect hair at regular intervals. For example, monitoring changes in hormonal levels as beavers age, or longitudinal diet studies based on stable isotope analysis of hair. Hair snares may not be the best approach if consistent samples are needed over time. Based on our methods for choosing deployment sites, we placed the majority of snares along beaver feeding trails because they were the most active beaver feature at ponds. Placing snares at feeding trails may therefore be the most efficient location, although our experimental design prevented us from testing this directly. The overrepresentation of hair snares along beaver feeding trails may explain the high standard error for all beaver feature variables in model 1 and the lack of significance. Further research may compare the efficiency of hair snares at beaver features in a more balanced way.

Previous research has indicated the use of suspended apples for baiting traps (Sobkowiak, Kochan & Kruszyński, 2021), but we found beaver chewed logs work. At all six snares baited with beaver chewed logs, remote videos captured beavers coming into contact with the snare while carrying the logs to the pond. However, samples were only collected at five of these snares, indicating contact at one snare was unsuccessful. Although aspen twigs are often used to bait beaver traps (Koenen et al., 2005), they did not appear to successfully bait hair snares in our study. At all 30 snares baited with aspen, the aspen was left on the snare until the leaves and sticks browned. Snares were baited with bare hands, potentially impacting the efficacy of the bait. We did not use castor as a bait because we did not want to introduce materials that may stick to the hair samples themselves (see discussion of potential cross-contamination below). However, depending on the intended use of the samples, further research may find castor effective as it is commonly used to live trap beavers in cage traps (Rosell & Kvinhaug, 2024).

Although snares targeted beavers the majority of the time, our video evidence indicated that other animals with visually similar hair come into contact with snares. Muskrats were the second most likely animal after beavers, which is not surprising given their habitat co-occurrence (Gauvin et al., 2020). Upon close visual inspection of beaver and muskrat specimens at the University of Minnesota Bell Museum Mammal Collection (2088 Larpenteur Ave W. St Paul, MN 55113), both muskrat and otter fur look nearly indistinguishable from beaver fur to the naked eye besides the length (beaver fur is generally longer). Use of hair snares in areas where these species overlap will therefore require either a tolerance for a small amount of other species’ hair in the analysis, or species identification using genetics (Foran, Minta & Heinemeyer, 1997) or microscopic approaches (Waits & Paetkau, 2005).

Although we did not record this as its own behavior, in many of the videos where beavers investigated the snare they also rubbed their cheeks up against barbed wire. This may have distributed scent onto the metal. Such behavior should be considered in the future for potential cross contamination to hair samples from scent secretions or saliva. Beavers chose to travel over and under snares in similar frequency and the side of the body that was snagged did not influence the amount of hair collected. If researchers aim to collect hair from the back or the belly for consistency, or reduce the chance of smaller animals such as muskrats from contacting the snare, snares should be adjusted up or down. Increasing the height of the snare will limit the sample size to the largest beavers, while decreasing the height will allow beavers of all sizes, including smaller yearlings and neonates to travel over the snare. However, beavers going over, under, or traveling alongside the snares seem to all be effective ways to collect samples.

Snares did not negatively impact beaver traveling or foraging behavior in 99.9% of videos, however beavers did investigate snares quite often. This indicates that beavers potentially do not change their travel patterns, increase their energy expenditure, decrease their foraging efficiency, or put themselves at greater risk of predation when snares are deployed. However, we did not test these factors directly and we want to point out that we did not test if any beavers developed an aversion to the snares despite snares having seemingly no impact on their ability to travel or forage. Future research should consider the potential for certain beavers to develop trap shyness or trap happiness through individual DNA identification from hair samples collected (Lincoln, Wirsing & Quinn, 2020), or changes in beaver response to hair snares depending on the time of year (Lamb, Walsh & Mowat, 2016).

As beavers are being translocated and reintroduced (Gaywood, 2018; Doden et al., 2022; Doden et al., 2023), following up on the genetic diversity of new populations is essential to ensuring the long term success of such restoration endeavors or intensive management methods (Taylor et al., 2024). Furthermore, hair snares provide a way to conduct mark-recapture studies on alive beavers without the use of expensive equipment such as cameras to identify unique tail patterns (Hinds et al., 2023). Collecting individual genetic IDs provides the opportunity to determine various population patterns including estimating density, population size, inbreeding, and carrying capacity. Furthermore, a growing range of techniques can be applied to low-quality DNA samples (Andrews et al., 2016; Hohenlohe, Funk & Rajora, 2021), increasing the utility of hair samples exposed to the elements while on the snare. Other hair snaring methods such as rub stations collect shed hair which provides lower DNA quantity than plucked hair (Monterroso et al., 2014), whereas barbed wire plucks the hairs providing a hair follicle and thus a higher quantity of DNA.

Conclusions

Our study provides in depth evidence of passive hair snare methods used to collect North American beaver hair and adds to a growing suite of hair snares used for a variety of wildlife monitoring projects (Bremner-Harrison et al., 2006; McDaniel et al., 2006; Long et al., 2007; Gardner et al., 2010; Henry & Russello, 2011; Hanke & Dickman, 2013; Patkó, Ujhegyi & Heltai, 2016; Lukacs et al., 2020). The ability to collect tissue samples without capture or handling offers clear and practical benefits by reducing stress and risk for both the study animal and researchers implementing sampling procedures. Furthermore, non-invasive methods are appealing to funding and permitting agencies, and minimizing risk to animals is often legally- mandated. Beyond safety and institutional benefits, many ecological practitioners believe that minimizing animal disturbance is an ethical obligation, which extends to the implementation of non-invasive methods when appropriate (Pauli et al., 2010; Zemanova, 2020). The combination of hair snares with newly developed computer assisted recognition of beaver species (Dytkowicz et al., 2024) will provide a powerful non-invasive tool to understand colonies with minimal disturbance in ways we have not yet been able to implement.

Supplemental Information

Supplemental Information 1 Hair samples from non-invasive hair snares for north American beavers (Castor canadensis)

Each data point provides information on individual hair samples.

Supplemental Information 2 Code for evaluating the efficacy of hair snares to non-invasively collect hair from north American beavers (Castor canadensis)

Code includes evaluation of hair collected from hair snares, which side of the body hair was collected from, beaver behavioral response to snares, the potential for bycatch at the snares.

Supplemental Information 3 Remote camera videos from hair snares placed around north American beaver (Castor canadensis) ponds

Each data point is an individual video of animals at hair snares.

Supplemental Information 4 Database of hair collected at hair snares placed around north American beaver (Castor canadensis) ponds

Each datapoint is a different hair snare deployed in northern Minnesota, USA between 2022 and 2023 to collect hair non-invasively from beavers.

Supplemental Information 5 Summary of each time a non-invasive hair snare was checked for samples

Each data point is a date summarizing when non-invasive hair snares placed around beaver (Castor canadensis) ponds in northern Minnesota, USA were checked for samples. Data includes if a hair sample was collected, what species it was from, and if the remote camera at the snare was checked.

We would like to thank Olivia Jensen, Sage Patchett, and Maeve Rogers who put in 1,000s of hours and miles in 2022 and 2023 deploying and maintaining hair snares and Dr. Thomas Gable, Austin Homkes and Dr. Ellen Candler who provided remote cameras. We also thank technicians on the Voyageurs Wolf Project in 2022 that assisted with snare deployment and maintenance that year (Izzy Evavold, Liv Coletta, Clara Dawson, Rose Newell, Rudi Boekschoten, and Mark Belew).

Additional Information and Declarations

Competing Interests

Author Contributions

Animal Ethics

Field Study Permissions

Data Availability

The authors declare there are no competing interests.

Dani R. Freund conceived and designed the experiments, performed the experiments, analyzed the data, prepared figures and/or tables, authored or reviewed drafts of the article, and approved the final draft.

Joseph K. Bump conceived and designed the experiments, authored or reviewed drafts of the article, and approved the final draft.

The following information was supplied relating to ethical approvals (i.e., approving body and any reference numbers):

We set out hair snares for this research per the University of Minnesota’s Institutional Animal Care and Use Committee (protocols: MWR_VOYA_WINDELS_WOLF and UMN protocol no. 2207-40241A).

The following information was supplied relating to field study approvals (i.e., approving body and any reference numbers):

Field work was approved by the State of Minnesota Department of Natural Resources Division of Fish and Wildlife (Special Permit No. 35003).

The following information was supplied regarding data availability:

The raw data and code are available in the Supplemental Files.

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
