# Peer review of "Evaluation of non-invasive hair snares for North American beavers (Castor canadensis): placement, efficiency, and beaver’s behavioral response"

_PeerJ, doi:10.7717/peerj.19080_

## Round 0.1 · original submission · Minor Revisions

Dear Dani Freund and Joseph Bump,
thanks for your contribution. Please consider the few comments of the reviewers and resubmit a revised version of the manuscript.
Yours
Clara Stefen

Reviewer 1 ·

Basic reporting

no comment

Experimental design

no comment

Validity of the findings

no comment

Additional comments

The authors have conducted a thorough investigation of hair snare placement, efficiency, and impacts on beaver behaviour. The study design is robust, with a large sample size of 64 hair snares deployed across 56 beaver ponds over two field seasons. This provides good statistical power and ecological replication. Another key strength of the study is the implementation of remote cameras to monitor beaver interactions with the snares is a key strength, allowing detailed behavioural observations. Overall, this is a well-designed and executed study that provides valuable insights into hair sampling methodology. The results provide practical guidance for researchers looking to implement this method and I consider this study a valuable contribution to the research community.

SPECIFIC COMMENTS:

Lines 85: Could the fact that the locations were within wolf territories affect beaver behaviour caught on cameras, i.e. the animals in the area are more cautious than in other locations?

Lines 112: Why were apples chosen as bait? Has it been reported in previously published literature that this might be an effective bait for beavers? Since you report that „No samples were collected from snares baited with apples.“, the bait choice requires some explanation.

Lines 148-155: The statistical analysis needs to be outlined in more detail, including any model selection criteria and significance thresholds used.

Lines 199-210: The manuscript would benefit from more information on the quality of DNA obtained from the hair samples. Were any genetic analyses attempted to assess sample viability? Even if not part of this study, a brief mention of potential DNA applications would be valuable.

The discussion section could be strengthened by comparing the findings to hair snaring methods for other species, and what implications this has for large-scale beaver monitoring programs.

Lines 214-215: Were beavers the most often captured species on cameras? If so, then this outcome can be explained just by beavers being more abundant in the study location than other other species.

Lines 260-263: The potential cross-contamination from scent marking on snares deserves more discussion – how might this impact genetic or hormonal analyses? Consider adding a sentence or two addressing this potential limitation and its implications for future studies.

Since you did not identify individual beavers, is it possible that some individuals learned to avoid snares? What would be the implications for practice?

Line 269: Consider adding a brief limitations section to the Discussion. This could address potential sources of bias or error, such as the inability to identify individual beavers, potential seasonal effects on hair shedding, or any spatial autocorrelation issues in sample collection.

·

Basic reporting

Line 81-82: It took me a minute to catch up with what is said here, but I think I got it. If I understand correctly, you're saying that you repeated 5 ponds between 2022 and 2023 so you would have a better chance of getting a fresh sample from that pond. I think adding just a short sentence on how your method was specifically to try to access different ponds each year unless something went awry would be helpful.
Line 84-85: I'm not certain you necessarily have to say "part of a different project", I think you could say the Voyageurs wolf project and cite anything about how wolves are eating beavers. For example, the chapter in The Wolf book that one of the authors of this paper was involved in writing.
Line 177: I think you might just need to add "times" after 4.09

Experimental design

I have some concerns with handling bait with bare hands, but there's not a whole lot you can do about that now.
The categorical dirt score is a really good idea.
Line 161-164: Grouping these areas together seems like a reasonable way to do this, particularly how it can be difficult to determine exact contact points through a trail camera video.

Validity of the findings

As a methods paper, the potential for this method to be used by others to collect beaver hair is a solid place to start with the ability to adjust based on the researchers' own objectives.

Additional comments

I think it could be useful to add just a sentence or two in the intro about the importance of studying beavers as a part of wolf diets. There are certainly studies in existence that show wolf populations can be sustained by prey that aren't ungulates (Roffler et al. 2023).

You should list each technician that helped in the acknowledgments.

·

Basic reporting

I found the reporting in this manuscript to be generally clear and professional, with some pretty minor grammatical suggestions provided below.

However, I have a few issues with the data reporting, and please see *** more major comments below.

1. The key results should be summarized probably in both a table, and a figure, not just littered all through the lengthy, somewhat wordy Results text. Results by baiting technique, by location, by season, etc.

The figures that are provided are excellent, and Figure 5 is wonderful.

2. Similarly, the statistical results must be shown, it is insufficient to just say you detected no statistical differences. And allowing the reader to see the numbers in a graph will help us discern patterns that may not have had adequate power to be detected statistically but are biologically important. And relatedly, it would help if the conceptual equation of what was actually being tested with the statistics were shown in Table 1. Were any data transformed? The description of the statistics is overall insufficient, remember, in theory a reader would be able to reproduce your results if they had the raw data. For example, it makes little sense to me why the model 1 and model 2 response variables are standardized by sample days? Why would sample days not be an explanatory variable?

L71. Suggest “as well as evaluating welfare”
L77-82. Could move this “boring” text to the Acknowledgements, but maybe this is specific to this tyle of journal
L86. Suggest using a semicolon after km2 instead of nested (())
L86. Mandatory comma after 2015 (2 independent clauses connected by a conjunction)
L118-119. Reword “from which side of the body hair is sample”
L127-129. Bit of a run-on, reword.
L163. “legs were sampled, because”
L174. “2023, respectively.”
L177. “4.09 times per day”
L178. Not completely clear as written. Does this mean 100-83% of the times they were other animals? Please clarify.
L207. This is super important, split out even if a short paragraph. And again, we need to see the stats.
L215. Mandatory comma “ species and we”
L217. Comma after e.g. – e.g.,
L218. “”(() “sample; Koren”

***218-222. This is an incredibly important finding. I think the authors should expand on it. The beavers didn’t avoid the snares, which means they didn’t change their paths, waste energy, miss out on food, or put themselves at greater risk of predation, for example.

L227-231. Awkward and long, please reword.
L234-236. Awkward
L238-239. Is there a word missing?
L258-259 “videos; however,” (always with this use of however)
L262. Unclear as written, later by subsequent other beaver individuals?
L280-281. “legally-mandated”
L292. “put in” is too colloquial

Experimental design

This research is well designed, given it was opportunistic.

L87. Can you please tell us how you know this? How are you defining a colony (very difficult in the west, could be very helpful).

L~116. Why not use castor scent? We do and it works great.

L159-161. It is unclear what the hypothesis is here, are you hypothesizing that dirt accumulates with sample time?

L226-227. Perhaps it is just the wording, but this seems like a mathematical impossibility.

Validity of the findings

See above.

The findings are valid, clear and compelling, given a few minor things that could be improved.

***L284-286. In addition to what is written, an incredibly important implication of this study is that people can use this technique to collect hairs for individual DNA, finally offering a way to do mark-recapture on tricky beavers. This estimation procedure will allow us to estimate density, population size, and carrying capacity. Please expand on this concept. People are translocation nuisance beavers all over the west (e.g., https://qcnr.usu.edu/beaver-restoration/) , in many cases with little evaluation of local beaver ecology making it all the more important currently to understand these population dynamics.

Doden, E. P. Budy, T. Avgar, and J.K. Young. 2022. Movement Patterns of Resident and Translocated Beavers at Multiple Spatiotemporal Scales in Desert Rivers. Frontiers in Conservation Science: Animal Behavior in Novel Environments 3:777797. DOI: 10.3389/978-2-88976-152-4.

Doden, E., P. Budy, M.M. Conner, and Julie K. Young. 2022. Comparing translocated beavers used as passive restoration tools to resident beavers in degraded desert rivers. Animal Conservation ISSN 1367-9430. doi:10.1111/acv.12846.

Sandbach, C. J.K. Young, M.M. Conner, E. Hansen, and P. Budy. 2024. Evaluating the Influence of Beaver Dam Analogs on Beaver Translocation for Desert River Restoration. Restoration Ecology. DOI: 10.1111/rec.14107.

Additional comments

Although straightforward and not complicated, the implications of this study are profound and will be of wide interest.

---

## Round 0.2 · accepted · Accept

Dear Dani Freud,

As you addressed the reviewers comments I am pleased to tell you that the last review recommended "accept" and your manuscript is accepted.

Reviewer 1 ·

Basic reporting

no comment

Experimental design

no comment

Validity of the findings

no comment

Additional comments

I am pleased to state that the authors have successfully addressed the comments and issues I had raised during the first round of the review process. These revisions have definitely strengthened their arguments and clarified the methods and results, making it easier for readers to follow and understand how the research was done as well as its importance. I have no further comments to add and congratulate the authors on a well-written and insightful study on non-invasive hair sampling in beavers.